



**An integrated method for assessing vulnerability of buildings caused by debris flows**
**in mountainous areas**
Chenchen Qiu[1], Xueyu Geng[1]*
[1]School of Engineering, University of Warwick, Coventry, CV4 7AL, UK
Xueyu Geng (**Corresponding Author**):
E-mail: xueyu.geng@warwick.ac.uk
School of Engineering, University of Warwick, Coventry, CV4 7AL, UK
**Abstract:** The vulnerability assessment of buildings in future scenarios is critical to decrease potential
losses caused by debris flows in mountainous areas due to the complex topographical condition that
could increase the environmental vulnerability to climate change. However, the lack of reliable methods
limits the accurate estimation of physical damage and the associated economic loss. Therefore, an
integrated method of physical vulnerability matrix and machine learning model was developed to benefit
the estimation of damage degree of buildings caused by a future debris-flow event. By considering the
building structures (reinforced-concrete (RC) frame and non-RC frame), spatial positions between
buildings and the debris-flow channels (horizontal distance (*HD*) and vertical distance (*VD*)), and impact
pressure ($P_t$) to buildings, a physical vulnerability matrix was proposed to link physical damage with the
four factors. In order to overcome the difficulty in estimating the possible impact pressure to buildings,
an ensemble machine learning (ML) model (XGBoost) was developed with the involvement of
geological factors. Additionally, the *HD* and *VD* were decided based on the satellite images. The
Longxihe Basin, Sichuan, China was selected as a case study. The results show that the ML model can
achieve a reliable impact pressure prediction because the mean absolute percentage error (MAPE), root
mean squared error (RMSE), and mean absolute error (MAE) values are 9.53%, 3.78 kPa, and 2.47 kPa.
Furthermore, 13.9% of buildings in the Longxihe Basin may suffer severe damage caused by a future



debris-flow event, and the highest economic loss is found in a residential building, reaching $5.1\times10^5$ €.
Overall, our work can provide scientific support for the site selection of future constructions.
**Keywords**: Debris flow, geological factors, building, machine learning, vulnerability assessment

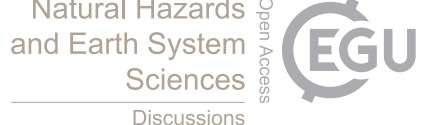

## 1. Introduction

Debris flows are among the most frequent and costly natural hazards due to climate change and
difficulty in timely warning (Santi et al., 2011). These events can devastate entire settlements in their
path and pose significant threat to natural environment (Immerzeel et al., 2020), causing destruction of
aquatic biodiversity, along with damage to properties and finally leading to considerable economic
losses worldwide each year (Qiu et al., 2022; Alene et al., 2024; Sridharan et al., 2024). In European
Alps, this disaster claimed an economic loss of at least 5 € billion from 1988 to 2012 (Fuchs, 2009;
Guzzetti et al., 2005). Moreover, a similar average annual loss is also found in China, approximately
0.17 € billion of annual loss was recorded during the time period of 2005 and 2015 (Miao and Liu, 2020).
In this case, a reliable estimation of the potential economic loss caused by debris flows is essential since
it can provide guidance for decision-makers about where to place the infrastructures and buildings. The
buildings are the most susceptible element to debris flows, and they are responsible for most of the
economic loss (Fuchs, 2009; Wei et al., 2018). Therefore, in order to calculate the potential economic
loss, it is critical to estimate the damage degree of the buildings since economic loss is linked to the
physical vulnerability of a property and its economic value.
The physical vulnerability quantifies the damage degree of a property, and the methods that are used
to decide the physical vulnerability include mechanical method (Ruggieri et al., 2023, 2022),
vulnerability matrices, vulnerability curves, vulnerability indicators (Papathoma-Köhle et al., 2017). The
mechanical methods derive the vulnerability functions of buildings based on numerical models, which
may achieve relatively high evaluation accuracy but highly rely on controlled laboratory experiments to
obtain input data. As a result, this method itself is time-consuming and costly for regional application
(Paudel et al., 2021; Qiu et al., 2022). Three vulnerability curves were derived using numerical
modelling to relate the vulnerability to debris-flow intensity, including flow height, flow velocity, and





kinematic viscosity (Quan Luna et al., 2011). Although these three curves can suggest the physical
vulnerability of a building at risk but fail to consider the impacts of building structures on damage
degree. Therefore, a brick structure and a reinforced-concrete frame were included in the development of
vulnerability curves by Zhang et al. (2018). However, the involvement of limited building types restricts
the application of the curves when the determination of physical vulnerabilities for different building
types is required. Therefore, considering the limitations of vulnerability curves, different vulnerability
matrixes of buildings have also been developed by many studies due its advantages in interaction
understanding between the debris-flow process and elements at risk and easily readable by non-experts
(Bründl et al., 2009; Kang and Kim, 2016; Zanchetta et al., 2004). In contrast, these developed matrixes
ignored the spatial position (horizontal distance and vertical distance) between the buildings and the
debris-flow channels, which would misestimate the damage degree of a building caused by a debris-flow
event. As for the vulnerability indicators, this method considers the characteristics of buildings without
relating the debris-flow process when evaluating the damage degrees (Fuchs et al., 2019). Therefore, it is
crucial to establish a comprehensive assessment matrix that takes into account the structural types,
spatial positions between buildings and the debris-flow channels, and debris-flow intensities to estimate
the potential damages of the buildings. Additionally, the possible damage degree of the buildings in
future scenarios was not considered by the past studies (Papathoma-Köhle et al., 2017). Therefore, this
study focuses on conducting an assessment of the potential physical damage of a building due to a future
debris-flow event.

Among the four factors in deciding the physical damage of buildings (building structure, spatial

locations ($HD$ and $VD$), and impact pressure ($P_t$)), impact pressure remains an unsolved problem since
$HD$ and $VD$ can be determined based on the satellite images. In this case, a machine learning model was
developed to predict the impact pressure to a building because this method can uncover intricate and





concealed relationships between various input variables and an output result (Khosravi et al., 2021; Jiang
et al., 2023). To leverage the benefits of rapid processing and handling large-scale data, we employ an
ensemble model, specifically extreme gradient boosting (XGBoost). This choice is made due to
XGBoost's ability to partition data into smaller components, facilitating parallel computation and
multithreading to enhance processing speed (Chen and Guestrin, 2016).

In this paper, we proposed an integrated method of physical vulnerability matrix and machine

learning model to estimate the physical damage of a building caused by a future debris-flow event,
finally estimating the economic loss of this property. The buildings in the Longxihe Basin, Sichuan,
Chian, were extracted to conduct a case study to test the efficiency and reliability of this method in
physical damage estimation and corresponding economic loss. The formation of terrain in this area is
affected by severe tectonic activities, which can produce abundant loose materials for potential debris
flows.
**2. Methodology**

To estimate the economic loss of buildings caused by a future debris-flow event, several steps are

comprised in this study (see Fig. 1):





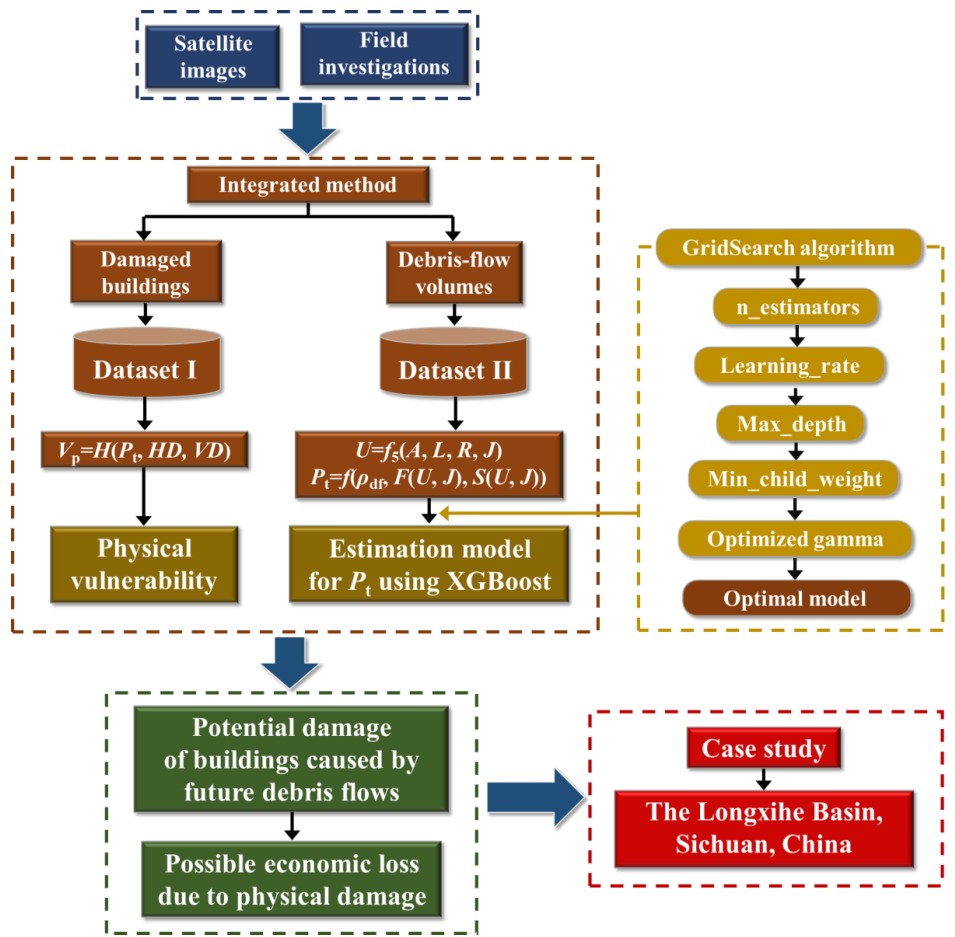


Figure 1. Flow chart of this study
(1) The historical debris-flow events in Gyirong, Tibet Tibetan Autonomous Region, and the
Sichuan Basin (Fig. 2) from the past ten years are investigated based on satellite images and field
investigations to collect information regarding the debris-flow volumes and damaged buildings.
(2) We categorize the collected historical debris flows into two datasets (dataset I and dataset II) for
the development of a physical vulnerability matrix and a prediction model, respectively.
(3) The dataset I includes the debris-flow events that caused damages to the buildings. Therefore,
this dataset is employed for the development of a physical vulnerability matrix.
(4) The dataset II is composed of the debris-flow events that occurred in areas without the


distribution of buildings, and, therefore, no property loss is caused by these events. Therefore, this
dataset was used for model training and utilize this model to estimate the debris-flow intensity in future
scenarios, such as debris-flow impact pressure to buildings. This dataset is shown in Table 6 of
Appendix. A.

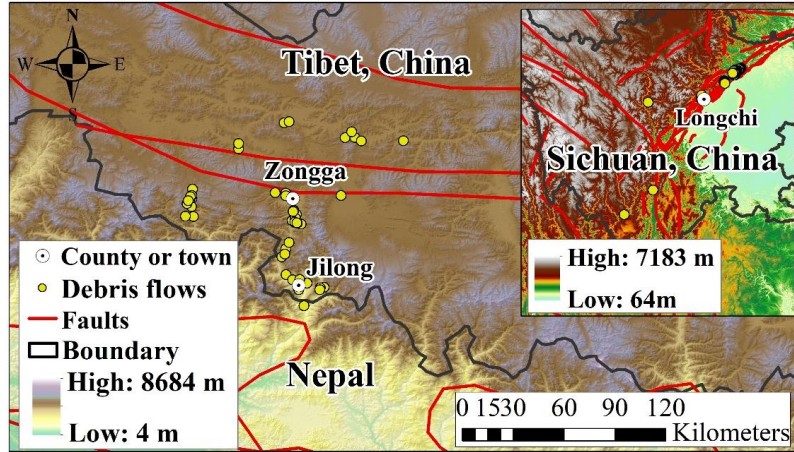


Figure 2. The collected historical debris flows in the Tibet Plateau and the Sichuan Basin
**2.1 Physical vulnerability matrix**
Vulnerability refers to the damage degree of a property when subjected to a hazard event, such as a
landside and a debris-flow event (Fell, 1994). Normally, a physical vulnerability was used to quantify
the damage degree of a property. To obtain the future economic loss of a building at risk, a physical
vulnerability matrix of the buildings was proposed. The determination of physical vulnerability relied on
the impact pressure ($P_t$) to buildings, the horizontal distance ($HD$), and the vertical distance ($VD$)
between the building and the nearest debris-flow channel, as indicated by Eq. (1). The determination
details of the three parameters in Eq. (1) are demonstrated in the following sections.
$$V_p = H\left(P_t, HD, VD\right)$$    (1)
**2.1.1 Calculation of impact pressure**
In order to propose a physical vulnerability matrix, the first step is to link the impact pressure to





damage degree. As suggested by (Jakob et al., 2012; Kang and Kim, 2016), $P_t$ can effectively reflect the
energy of debris flows and possible damage degree of buildings. However, past studies usually utilized
debris-flow magnitude to decide the physical vulnerability since a greater magnitude may indicate a
more significant impact force (Dai et al., 2002). This impact force cannot represent the actual damage of
a building during a debris-flow event because the catchment with a potential large-scale debris-flow
event may not cause severe damage to the buildings. The reason behind this uncertainty could be due to
the moderate slope gradient and frictional resistance of a debris-flow channel, which could decrease the
kinetic energy of the travelling mass. Consequently, only a slight or moderate damage to buildings could
be caused. Therefore, impact pressure can better reflect the damage degree of buildings when subjected
to different debris-flow magnitudes, which can be calculated through considering the dynamic
overpressure and hydrostatic pressure (Eq. (2)) (Zanchetta et al., 2004):

$$P_t = \frac{1}{2}\rho_{df}\,gh + \rho_{df}\,v^2 = f\left(\rho_{df}, h, v\right) \tag{2}$$

where $P_t$ (kPa) represents the impact pressure to buildings, and $g$ is the gravitational acceleration. $v$ (m/s)
represents the mean flow velocity, and $\rho_{df}$ is the mean density of materials for a debris-flow event. $h$ (m)
is the flow depth. As for the debris-flow velocity ($v$) at peak discharge, it can be calculated using the
equation proposed by Rickenmann (1999). This equation considers the debris-flow datasets in different
regions, such as Italy, China, Japan, U.S.A, and Columbia, which enables its feasibility to be used in
wider and different areas.

$$v = 2.1 Q_p^{0.33} J^{0.33} = f_1\left(Q_p, J\right) \tag{3}$$

This equation illustrates that the velocity can be decided by $Q_p$ (m³/s) and channel gradient ($J$) (Cui
et al., 2013). The calculation of $Q_p$ can be determined based on the equation (Eq. (4)):

$$Q_p = \left(U/152.97\right)^{1/1.266} = f_2\left(U\right) \tag{4}$$

Therefore, the $Q_p$ can be calculated based on the estimated volume ($U$ (m³)) of historical debris


flows. However, the absence of flow depth ($h$) also hampers the calculation of impact pressure.
Therefore, an equation is used to calculate the flow depth (Koch, 1998). This formula has been proven to
perform well in the numerical simulation of viscose debris flows (Eq. (5)):
$$h = \left( v / C_1 J^{0.5} \right)^{10/3} = f_3 \left( v, C_1, J \right) = f_3 \left( f_1 \left( Q_p, J \right), C_1, J \right) \tag{5}$$

where $C_1$ represents the dimensional empirical coefficient. This value of parameter is indicated by a
semi-theoretical relationship (Eq. (6)) (Rickenmann, 1999):
$$C_1 = 10 Q_p^{2/25} = f_4 \left( Q_p \right) = f_4 \left( f_2 \left( U \right) \right) \tag{6}$$

Therefore, the impact pressure can be described as a function of debris-flow volume and channel

gradient, and the impact pressures of dataset I are calculated based on Eqs. (2)-(6) (see Table 1).
**2.1.2 Determination of *HD* and *VD* values**

*HD* and *VD* values were also introduced here since the actual damage will be significant if a

building stands close to the debris-flow channel (Sturm et al., 2018). They can be estimated through
high-resolution satellite images, such as Gaofen, Ziyuan, WorldView, and GeoEye. In this study,
Gaofen-2 satellite images are employed for determining the *HD* and *VD* values. This satellite can
capture panchromatic (black and white) images with a spatial resolution reaching 0.8 m and
multispectral (color) images with a spatial resolution up to 3.2 m. Therefore, the resolution of satellite
images used for buildings identification is 0.8 m. As for the building clusters that are hard to be
separated into individual buildings manually, a 'fishnet' tool in GIS was used to automatically divide
these clusters into building segments. Furthermore, the rectangle segments were converted into points so
that each point represents a building. As a result, the *HD* and *VD* values of a building can be decided.
The damaged buildings are mainly distributed on the accumulation fans. Therefore, even though a large
*HD* is observed, the *VD* is small due to the mild slope and smooth topography of the alluvial fans
(Marcato et al., 2012). By considering the impact pressure, *HD*, and *VD* values, a physical vulnerability





matrix can be established to evaluate the physical damage of a building caused by a debris-flow event.

**2.2 Economic loss of a building at risk**

The economic loss of a building caused by a debris-flow event can be estimated based on
multiplication of its physical vulnerability and economic value.

$$V_\mathrm{e} = V_\mathrm{p} \times M = H\left(P_\mathrm{t}, HD, VD\right) \times M; M = P \times A \tag{7}$$

where, $V_\mathrm{e}$ and $M$ represent the economic loss and the economic value of a building, respectively. $P$ is the
unit price of a building, and $A$ represents the area of a building. Therefore, estimating $V_\mathrm{p}$ holds
paramount importance in estimating economic loss. However, $V_\mathrm{p}$ ($H(P_\mathrm{t}, HD, VD)$) is represented by the
proposed physical vulnerability matrix. In this context, determining $P_\mathrm{t}$ plays a critical role in economic
loss estimation. Therefore, to forecast the possible economic loss caused by a future debris-flow event,
we need to estimate the impact pressure to buildings caused by a future debris-flow event.

**2.3 Prediction model development**

To predict the future impact pressure to buildings when a debris-flow event occurs, determining
factors is essential. Therefore, we further developed Eq. (5) by integrating Eq. (4) and Eq. (6) to this
equation:

$$h = f_3\left(f_1\left(f_2\left(U\right), J\right), f_4\left(f_2\left(U\right)\right), J\right) = F\left(U, J\right) \tag{8}$$

Additionally, Eq. (3) can be rewritten as:

$$v = f_1\left(Q_p, J\right) = f_1\left(f_2\left(U\right), J\right) = S\left(U, J\right) \tag{9}$$

Therfore, the determination of impact pressure reslies on $U$ and $J$:

$$P_\mathrm{t} = f\left(\rho_{df}, F\left(U, J\right), S\left(U, J\right)\right) \tag{10}$$

However, the debris-flow volume is closely related to a set of geomorphic factors, as suggested by
Huang et al. (2020). They are catchment area ($A$), channel length ($L$), topographic relief ($R$), and mean
slope of the main channel ($J$). The catchment area can reflect the debris availability and capacity of



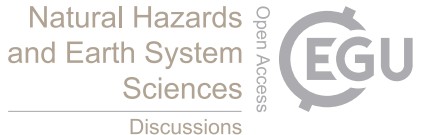

generating and containing the volume of loose materials for a debris-flow catchment. As for the channel
length, it is related to the entrained and transported sediment volume (Marchi et al., 2019). Therefore,
this parameter can also impact the final volume of a debris-flow event. $R$ is defined as the terrain
fluctuation and roughness of a catchment. To calculate this value, we need to first decide the optimal
statistical unit in this area using the change-point model. Then, the subtraction value between the
maximum value and minimum values of an optimal statistical unit is calculated. Finally, we utilized the
maximum subtraction value to represent the $R$ value of a catchment. $J$ is defined as the ratio of the
elevation difference of the main channel and channel length. A longer distance could be achieved for a
debris-flow event if a steep channel exists in a catchment (de Haas and Densmore, 2019). In this case, $U$
can be described as a function of $A$, $L$, $R$, and $J$:

$$U = f_5(A, L, R, J) \tag{11}$$


Furthermore, subsitituting Eq. (11) to Eq. (10):

$$P_t = f\left(\rho_{df}, F\left(f_5(A, L, R, J), J\right), S\left(f_5(A, L, R, J), J\right)\right) \tag{12}$$


Therefore, $P_t$ can be described as a complex function of geomorphology-related factors, including $A$,
$L$, $R$, and $J$. To find the complicated correlations among them, an ensemble machine learning model
(extreme gradient boosting (XGBoost)) was employed here to establish the relationship and then utilize
this relationship to estimate the potential impact pressure to buildings when a future debris-flow event
occurs. The basic mechanism of XGBoost is to constantly develop a new decision tree which acts as a
weak learner and fits the residual error of the last prediction. After the training of a total of $k$ trees, the
final prediction result is the sum of the score of each leaf node in each developed tree. Overall, the target
function of regression is placed in Appendix. A. Additionally, the database II that is used for impact
pressure prediction is presented in Table 6 of Appendix. A.
**2.4 Model assessment**


After the impact pressure prediction, three assessment indexes were used to evaluate the prediction
performance, including MAPE (Mean Absolute Percentage Error), RMSE (Root Mean Square Error),
and MAE (Mean Absolute Error):
$$MAPE = \frac{1}{m}\sum_{i=1}^{m}\frac{\left|y_i - y_{ipre}\right|}{y_i} \tag{13}$$

$$RMSE = \sqrt{\frac{1}{m}\sum_{i=1}^{m}\left(y_i - y_{ipre}\right)^2} \tag{14}$$

$$MAE = \frac{1}{m}\sum_{i=1}^{m}\left|y_i - y_{ipre}\right| \tag{15}$$

where $y_i$ is the actual value, and $y_{ire}$ represents the prediction value. $m$ is the number of prediction values.
**3. Result analysis**
**3.1 The relationship between the damage degree and $P_t$**
Fig. 3 shows the different damage degrees of buildings in dataset I. The buildings were classified
into two types, including RC-frame (reinforced concrete) and non-RC frame (masonry, wooden structure,
and light steel frame). As indicated in Figs. 3(e)-(f), The masonry buildings suffer severe damage, and
the light steel frame buildings and wooden structure buildings are destroyed (Figs. 3(g)-(h)) even though
the impact pressure to buildings was estimated to be less than 30 kPa. However, the main structure of
the reinforced concrete building can stay undamaged (Fig. 3(b)) when severe damage is found on the
masonry structure (see a dashed circle in Fig. 3(b)) during the same debris-flow event. This resistance
ability difference indicates the difference in physical vulnerabilities between the RC and the non-RC
frames, which can also be seen in Fig. 3(a). Moreover, moderate damage to the RC frame with
unreinforced masonry infill walls is found in Fig. 3(c) when a small-scale debris-flow event occurs.
Additionally, the RC frame suffers extensive damage when the impact pressure exceeds 100 kPa based
on the estimated debris-flow volume. Therefore, the identifications of different damage degrees for
buildings provide us with access to proposing a classification standard for the physical vulnerability of



buildings.

Figure 3. Photographs of the damaged residential buildings caused by debris flows during the field





investigations on the Qinghai-Tibetan Plateau

**3.2 Determination of *HD* and *VD* thresholds**

The field investigations and statistical results show that the non-RC frame buildings are destroyed or

suffer structural damage when the *HD* is less than 30 m (Fig. 4(a)). The damaged buildings cannot be

repaired, and reconstruction is required. In consistent with the conclusion of past study (Wei et al., 2022),

the residential buildings, such as brick structures (Fig. 4(b)) and the RC frame buildings (Fig. 4(c)), are

partially buried by the debris-flow sediments without structural damage when the *HD* is greater than 100

m but less than 160 m. Therefore, 160 m is another *HD* threshold to classify the inundated and slightly

affected areas. The upper limit of *HD* value for the historical debris flows during the field investigations

is 230 m because almost 94% of *HD* values are less than 230 m (see Table 1).

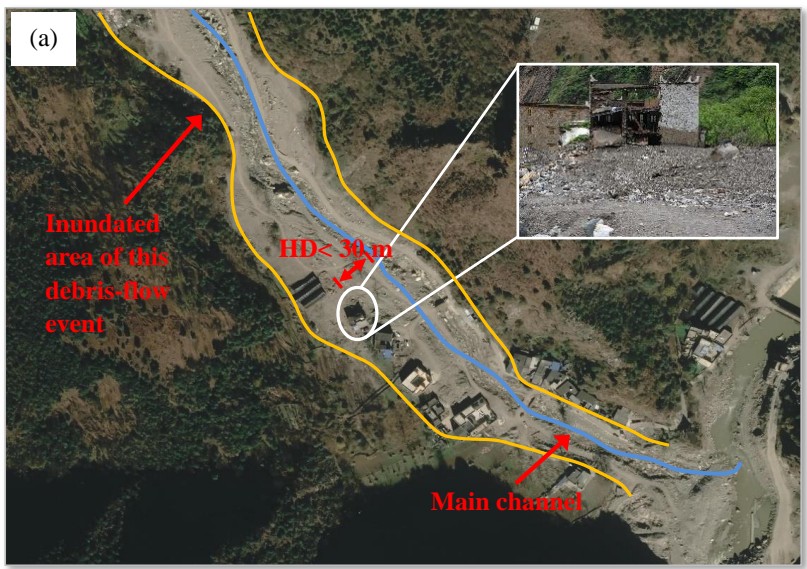





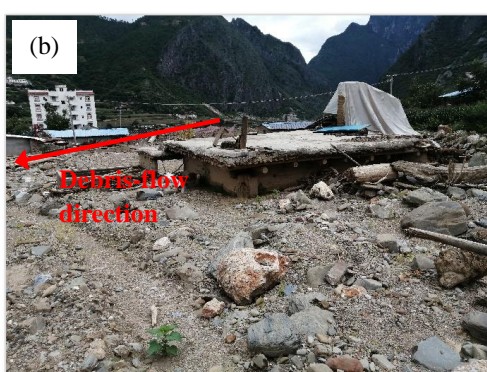 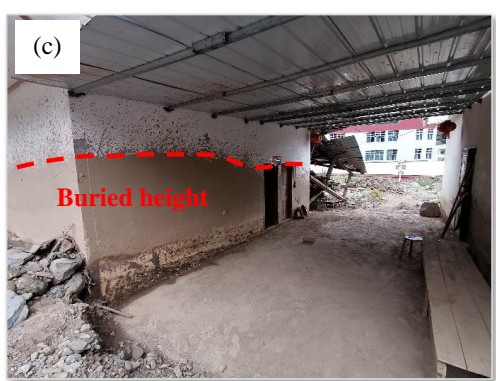

Figure 4. Examples of the determination of the *HD* thresholds

**Table 1**. Dataset I for physical vulnerability matrix.

| No. | Year | Lon (°) | Lat (°) | Number of damaged buildings | Impact pressure $P_t$ (kPa) | Maximum *HD* (m) | Maximum *VD* (m) |
|---|---|---|---|---|---|---|---|
| 1 | 2006 | 85.3278 | 28.3735 | 21 | 16.1 | 162 | 12 |
| 2 | 2007 | 85.5683 | 29.1875 | 13 | 40.6 | 141 | 12 |
| 3 | 2007 | 85.5528 | 28.8717 | 7 | 37.5 | 13 | 7 |
| 4 | 2008 | 85.6241 | 29.1869 | 21 | 41.0 | 119 | 3 |
| 5 | 2010 | 86.0872 | 29.1625 | 11 | 35.5 | 54 | 2 |
| 6 | 2013 | 85.3112 | 28.7649 | 53 | 24.1 | 284 | 29 |
| 7 | 2015 | 85.2928 | 28.4174 | 9 | 117.4 | 160 | 2 |
| 8 | 2015 | 85.3608 | 28.4074 | 22 | 31.1 | 131 | 107 |
| 9 | 2015 | 85.3542 | 28.7159 | 7 | 17.5 | 82 | 13 |
| 10 | 2015 | 84.7653 | 28.7559 | 38 | 132 | 74 | 15 |
| 11 | 2015 | 85.4566 | 28.3868 | 3 | 5.1 | 32 | 10 |
| 12 | 2015 | 85.4413 | 28.3827 | 1 | 32.7 | 17 | 6 |
| 13 | 2015 | 85.0105 | 29.1208 | 3 | 5.2 | 133 | 2 |
| 14 | 2015 | 85.2579 | 29.2603 | 9 | 9.8 | 146 | 2 |
| 15 | 2015 | 85.2759 | 29.2652 | 6 | 14.8 | 228 | 10 |
| 16 | 2015 | 85.0083 | 29.1493 | 4 | 14.6 | 171 | 3 |

In order to support the thresholds determination of *HD*, we further analyzed the frequencies of *HD*

values for the damaged buildings, as depicted in Table 1, through dividing the *HD* values into several

intervals. The frequency and accumulative frequency results are shown in Fig. 5.


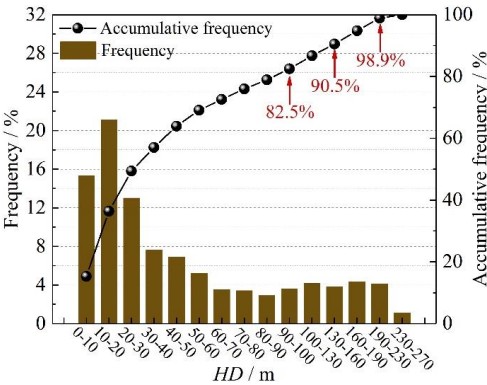


Figure 5. The frequency and accumulative frequency distributions of the 228 damaged buildings.


As depicted in Fig. 5, the highest proportion occurs in the range of 10 to 20 m, accounting for 20.1%,
followed by a 15% percentage of *HD* values falling between 20 to 30 m. Therefore, the proportion
falling within the range of 0 to 30 m is 49.4%, and approximately 82.5% of the *HD* values is measured
under 100 m. Following the suggestion of Liu et al. (2020), a probability of 50% is considered a
threshold for debris-flow warning, which implies that 30 m in this study can serve as a threshold.
Moreover, the accumulative frequency of 80% is selected as another threshold based on Wei et al. (2018),
corresponding to the *HD* value of 100 m. Furthermore, 90.5% of the damaged buildings have *HD* value
less than 160 m, and nearly 98.9% of the damaged buildings fall within the *HD* range of 0 to 230 m. As
a result, 160 m and 230 m are selected as additional two thresholds. In addition to the determination of
*HD* threshold values, the maximum flow depth (*h*) in the debris-flow channel is used as a reference to
decide the *VD* thresholds since the buildings are mostly situated along the channels (Fig. 4(a) and Fig. 6).
Therefore, calculating the elevation difference between the buildings and the nearest debris-flow
channel is critical to evaluate the safety of the buildings. For example, both the masonry buildings in Fig.
4(a) and Fig. 6 are close to the debris-flow channel. However, no severe damage is observed for the
building in Fig. 6 because it has a considerable vertical distance from the main channel. To decide the
*VD* thresholds, the *h* values of the historical debris flows are presented in Table 6 of Appendix. A. The





average depth of the debris flows is 2.6 m, and nearly all the *VD* values are less than 4 m. Therefore, 4 m
serves as the first threshold, suggesting that the most severe damage to the buildings may be caused
when the *VD* is less than 4 m. Whilst a debris-flow depth value of as high as 10 m is suggested (Xie et
al., 2013), which can be found in curved channels. Consequently, we utilize 10 m to indicate the
moderate damage of buildings when the *VD* is less than 10 m but greater than 4 m. Moreover, a vertical
distance of 14 m above the river level is considered to record the river gauging on the Iowa River using a
digital video camera (Creutin et al., 2003), which indicates a safe *VD* value to avoid damage caused by
the river discharge. Therefore, 15 m is used as the upper limit of the *VD* values in this paper.

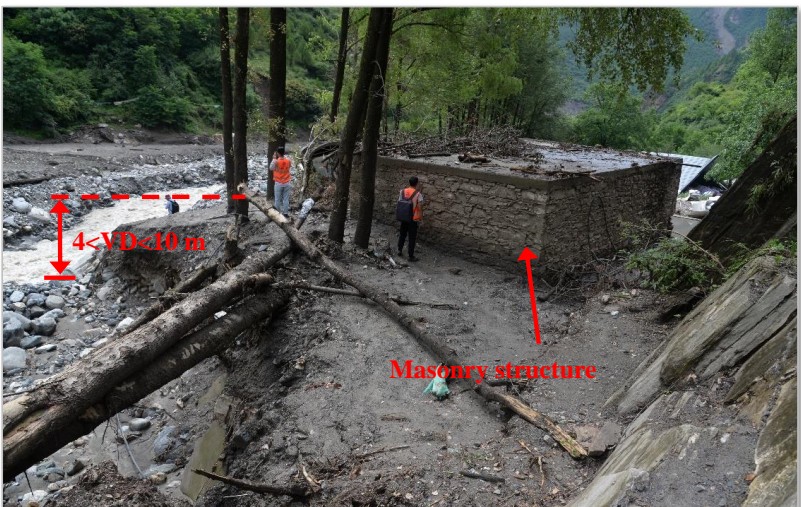


Figure 6. Example of the determination of the *VD* threshold

**3.3 Physical vulnerability matrix ($h(P_t, HD, VD)$)**

The proposed physical vulnerabilities of residential buildings are listed in Table 2. Extensive

damage or even complete damage may occur when a non-RC building is located near the debris-flow
channel with *HD* less than 30 m and *VD* less than 4 m. However, a significant improvement in resistance
ability can be observed when the non-RC frame is replaced by the RC frame considering the same
impact pressure, *HD* and *VD* values. In general, the buildings hardly suffer damage when the *VD* is





greater than 10 m. Therefore, the economic loss of a building can be calculated based on the proposed
physical vulnerabilities and economic values.

Table 2. Physical vulnerability matrix

| $P_t$ (kPa) | Building structure | $HD<30$ m | | | $30<HD<100$ m | | |
|---|---|---|---|---|---|---|---|
| | | $4<VD$ | $4<VD<10$ | $10<VD<15$ | $4<VD$ | $4<VD<10$ | $10<VD<15$ |
| <30 | RC frame | 0.3 | 0.2 | 0.1 | 0.2 | 0.1 | / |
| | Non-RC frame | 0.8 | 0.7 | 0.6 | 0.7 | 0.6 | 0.4 |
| 30-70 | RC frame | 0.6 | 0.5 | 0.4 | 0.5 | 0.4 | 0.2 |
| | Non-RC frame | 1 | 0.9 | 0.8 | 0.9 | 0.8 | 0.6 |
| 70-100 | RC frame | 0.7 | 0.6 | 0.5 | 0.6 | 0.5 | 0.3 |
| | Non-RC frame | 1 | 1 | 0.9 | 1 | 0.9 | 0.7 |
| >100 | RC frame | 0.8 | 0.7 | 0.6 | 0.7 | 0.6 | 0.4 |
| | Non-RC frame | 1 | 1 | 0.9 | 1 | 1 | 0.8 |
| $P_t$ (kPa) | Building structure | $100<HD<160$ m | | | $160<HD<230$ m | | |
| | | $4<VD$ | $4<VD<10$ | $10<VD<15$ | $4<VD$ | $4<VD<10$ | $10<VD<15$ |
| <30 | RC frame | 0.1 | / | / | / | / | / |
| | Non-RC frame | 0.6 | 0.4 | 0.1 | 0.4 | 0.1 | / |
| 30-70 | RC frame | 0.4 | 0.2 | / | 0.2 | / | / |
| | Non-RC frame | 0.8 | 0.6 | 0.3 | 0.6 | 0.3 | / |
| 70-100 | RC frame | 0.5 | 0.3 | / | 0.3 | / | / |
| | Non-RC frame | 0.9 | 0.7 | 0.4 | 0.7 | 0.4 | / |
| >100 | RC frame | 0.6 | 0.4 | 0.1 | 0.4 | 0.1 | / |
| | Non-RC frame | 1 | 0.8 | 0.5 | 0.8 | 0.5 | 0.1 |

**3.4 Prediction model development and assessment**
The debris flows in Table 6 (see Appendix. A) were divided into a training set and a validation set,
and the training set is used to train the prediction model. The validation results are plotted in Fig. 7.
Additionally, the performance of the developed model is assessed using the three indexes (Eqs. (14)-
(16)). As indicated in Fig. 7, the prediction results show minor errors to the actual values, and the MAPE,
RMSE and MAE values are 9.70%, 3.98 kPa and 2.74 kPa, respectively. RMSE value can reflect the
extreme errors, and the calculated RMSE value can indicate that there are no extreme values observed in



the prediction results. Additionally, MAPE reflects the error percentage between the measured and
predicted values, and the model is more reliable if the MAPE is closer to 0. Therefore, it can be
concluded that this model performed well in predicting the volume of a future debris-flow event.

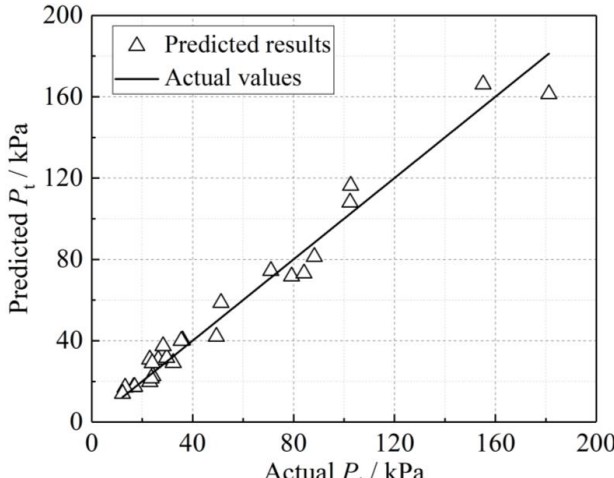


Figure 7. Plotted results of the prediction results

**4 Case study**
**4.1 Geological setting**

We selected the Longxihe Basin (Fig. 8) in Dujiangyan, Sichuan Province, to conduct a case study

(see Fig. 1 about the geographic location of this area), which is 15 km away from the epicenter of the
2008 Wenchuan earthquake. There are three faults crossing this area, namely the Southern Branch of the
Yingxiu-Beichuan Fault, the Northern Branch of the Yingxiu-Beichuan Fault, and the Feilaifeng
Structure. These faults and structures cause the incised valleys and uplifting of the land surface, resulting
in large areas of exposed rocks. Additionally, this study area belongs to the subtropical monsoon climate,
with annual precipitation reaching 1134.8 mm. Over 80 % of the annual rainfall occurs from May to
September. Consequently, the abundant rainfall and complex geological structure give birth to frequent
debris flows. It was reported that 13 debris-flow events occurred in this basin on 12[th] May, 24[th] June,
25[th] September 2008, and 17[th] July 2009. In particular, 45 debris-flow events were recorded on 13[th]





August 2010 due to a high-intense rainfall event, causing severe damage to 233 buildings and resulting
in the entire economic loss of $7.2 \times 10^7$ €. There are one town and two villages distributed in this basin.
The impacts of the Wenchuan earthquake still pose threats to the local people since a time period of at
least 20 years is required if the occurrence frequency of debris flows before the earthquake is expected
(Yu et al., 2014).

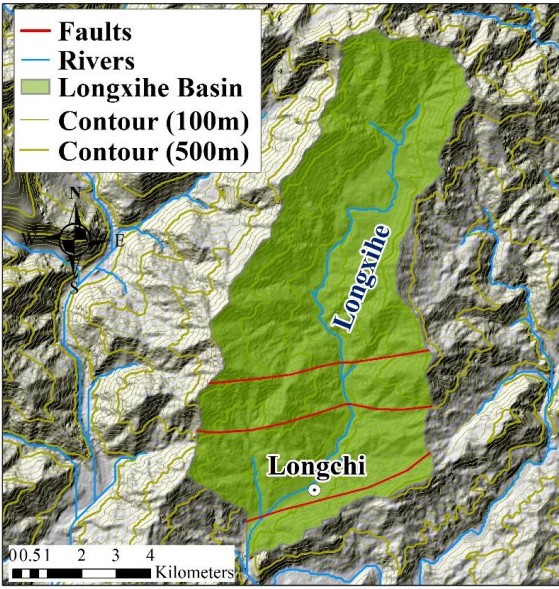


Figure 8. The Longxihe Basin, Dujiangyan, China
**4.2 Estimation of economic loss of buildings**
**4.2.1 Determination of physical vulnerability**
To estimate the potential physical damage of the buildings in the Longxihe Basin, the developed
prediction model was applied to predict the potential impact pressure to buildings. As illustrated in Fig.
9(c), the debris-flow catchments within this basin were generated since we mainly focus on the regions
with the distribution of buildings and estimate the possible economic loss of the buildings when debris
flows occur. Therefore, we extracted a total of 386 buildings in three regions based on the satellite
images (Fig. 9(a), Fig. 9(b), Fig. 9(d), and Fig. 9(e)). After that, we selected the catchments that are the





nearest to the buildings to conduct analysis (see highlighted catchments with red lines in Fig. 9(c)). The
input information of these catchments for impact pressure prediction and the predicted results are all
listed in Table 3.

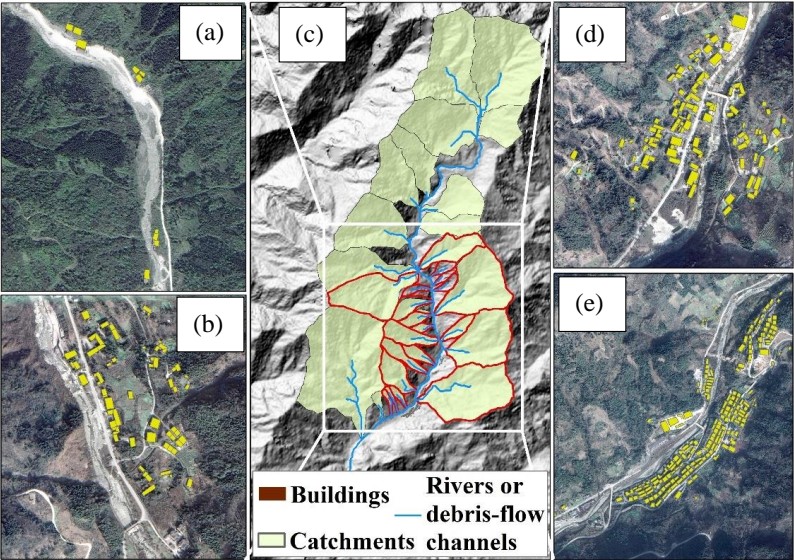


Figure 9. The debris-flow catchments and residential buildings in this area

Table 3. Prediction results using developed prediction model

| No. | $A$ / km$^2$ | $L$ / km | $R$ / m | $J$ | Predicted $P_t$ (kPa) |
|---|---|---|---|---|---|
| 1 | 0.4226 | 0.70 | 116 | 0.3024 | 22.0 |
| 2 | 0.8849 | 1.00 | 123 | 0.3503 | 26.7 |
| 3 | 0.1447 | 0.25 | 113 | 0.4055 | 18.0 |
| 4 | 2.9068 | 0.91 | 145 | 0.1668 | 22.1 |
| 5 | 0.3637 | 0.58 | 125 | 0.2998 | 19.2 |
| 6 | 0.9317 | 0.88 | 130 | 0.2551 | 20.9 |
| 7 | 4.1780 | 1.84 | 141 | 0.0751 | 16.0 |
| 8 | 0.1632 | 0.61 | 117 | 0.3419 | 19.3 |
| 9 | 0.0932 | 0.69 | 112 | 0.3622 | 17.3 |
| 10 | 0.1087 | 0.69 | 112 | 0.3542 | 17.5 |
| 11 | 0.2355 | 0.73 | 159 | 0.6828 | 16.5 |
| 12 | 1.3027 | 1.46 | 145 | 0.3944 | 25.2 |
| 13 | 2.8095 | 1.30 | 158 | 0.2466 | 26.5 |
| 14 | 0.3802 | 0.89 | 129 | 0.4299 | 19.2 |
| 15 | 0.2177 | 0.70 | 136 | 0.5690 | 15.8 |
| 16 | 0.1529 | 0.84 | 162 | 0.6821 | 14.4 |
| 17 | 3.5789 | 2.23 | 153 | 0.3047 | 33.6 |
| 18 | 0.3179 | 0.69 | 127 | 0.5400 | 17.4 |
| 19 | 0.1970 | 0.74 | 96 | 0.4056 | 15.0 |
| 20 | 0.2201 | 0.90 | 110 | 0.4599 | 13.0 |



334  In addition to the predicted impact pressures to the buildings by the potential debris flows, the

335 horizontal and vertical distances between each building and the nearest debris-flow channel were

336 measured using GIS. As a result, the physical vulnerabilities of the buildings in Longxihe Basin can be

337 decided based on the proposed physical vulnerability matrix, and the results are shown in Figs. 10(a)-(d).

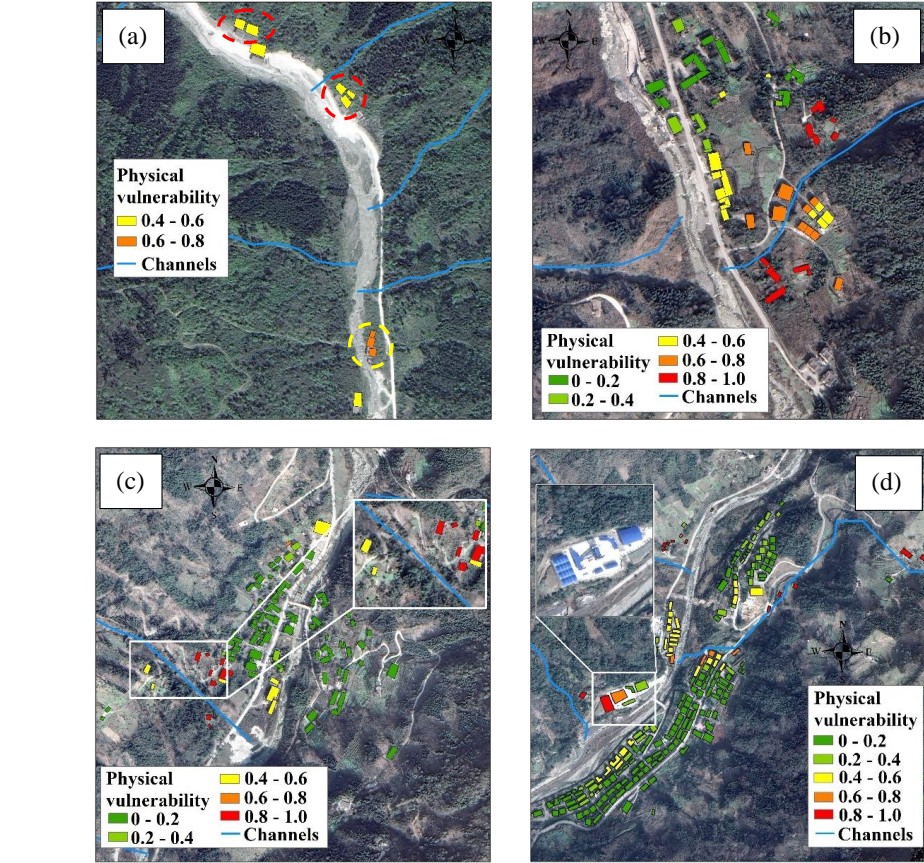

340        Figure 10. Physical vulnerabilities of the buildings in the Longchi Basin

341       Table 4. Statistical results of the buildings with different physical vulnerabilities

| | 0 - 0.2 | 0.2 - 0.4 | 0.4 - 0.6 | 0.6 - 0.8 | 0.8 - 1.0 |
|---|---|---|---|---|---|
| Number | 237 | 52 | 45 | 18 | 34 |
| Percentage | 61.4% | 13.5% | 11.6% | 4.7% | 8.8% |

342  The statistical results in Table 4 illustrate that most buildings nearly suffer no damage when a

343 debris-flow event occurs. This is because these buildings are RC-frame structures, which allow them to

344 stay undamaged or only suffer slight damage even though they are close to the debris-flow channels.



However, non-RC frame buildings may always suffer severe damage during a debris-flow event if their
locations are near the channels. As indicated in Figs. 10(a)-(d), the buildings with high and very-high
physical vulnerabilities are mainly brick and light steel structures. The difference in resistance ability
allows a greater possibility for RC-frame buildings to keep structures undamaged during the same
debris-flow event when compared to a non-RC building, which is consistent with the field investigation
results in Fig. 3(b). Moreover, a non-RC frame building can also avoid damage even though it is close to
the debris-flow channel. This is because a higher vertical distance to the debris-flow channel can allow
this non-RC building to suffer no damage or light damage. Therefore, a comprehensive analysis by
considering the structure type, spatial distances to debris-flow channel, and potential impact pressure is
significant to establish a reliable physical vulnerability matrix to benefit the determination of the
potential damage degree of buildings.
In order to validate the efficiency and accuracy of our method in estimating the physical damages of
buildings, the damaged buildings caused by debris flows on 13th August 2010 are employed here to
assess the reliability of this method. As depicted in Fig. 11(a), the RC-frame buildings suffer a moderate
damage (see red dashed circles in Fig. 10(a)) because there are no obvious damages of external or
internal walls observed during the field investigations based on the HAZUS building classification
scheme (Rojahn, 1988). However, the debris-flow event caused an extensive damage (see yellow dashed
circles in Fig. 10(a)) to the brick structures due to the partly destroyed external or internal walls. As a
result, evacuation of people is necessary and reconstruction is required. Overall, our proposed method
can provide a reliable evaluation of physical vulnerability of buildings caused by a debris-flow event
and therefore benefit their estimation of economic loss.


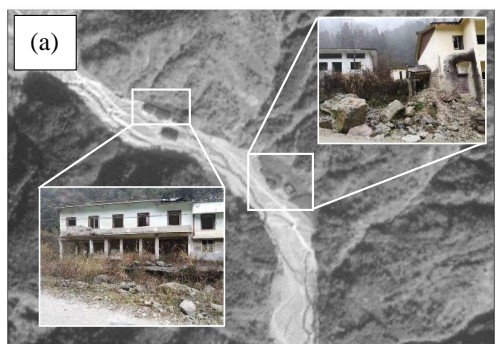 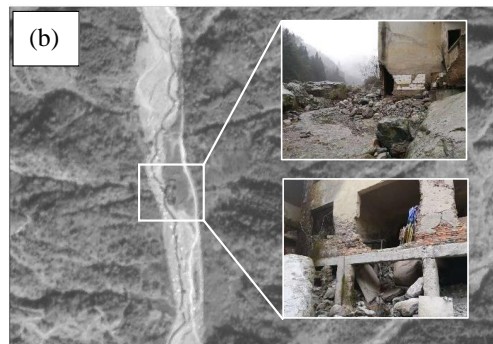

Figure 11. The damaged buildings caused by the debris-flow events.

**4.2.2 Economic loss**

Based on the estimated physical damage, we can further provide a reliable estimation of the economic loss. Six categories of buildings were identified in this study area based on the field investigations. They are residential buildings, factory buildings, office buildings, and livestock houses.

Table 5. Unit price ($P$) of a building in this area

| Element | Categories | Unit price | Value based on |
|---|---|---|---|
| Buildings | Residential buildings (RC-frame) | 1050.44 €/m² | Average market price |
| | Residential buildings (Brick structure) | 158.38 €/m² | Construction cost |
| | Business buildings (RC-frame) | 1371.47 €/m² | Average market price |
| | Office buildings (RC-frame) | 1050.44 €/m² | \ |
| | Factory buildings (Light steel structure) | 237.57 €/m² | Construction cost |
| | Livestock houses (Brick structure) | 7.92 €/m² | Restoration and reconstruction cost |

The economic value of a residential building in this area is based on the market price, which is provided by the Housing and Urban-rural Construction Agency. As for the unit price of a business building, we refer to the price ratio of a residential building and a business building in the city center of Dujiangyan. The unit price of a business building is normally 30% higher than a residential building. An office building belongs to the national assets, which cannot be rented or sold. However, possible damage still cannot be avoided if a debris-flow event occurs, which therefore requires restoration or





reconstruction. Therefore, we refer to the unit price of a residential building to estimate the economic
loss of an office building. Unlike the high construction cost and business value of a residential building
and a business building, the construction cost of a factory building is low because of its light steel
structure. Meanwhile, this kind of building is normally situated at a distance from the city center and
residential areas, primarily to mitigate effects of noise and environmental pollution. Most importantly, a
factory building invariably occupies a large area, potentially raising the construction cost when situated
in the city center due to the exorbitant land prices. Considering the average market price of a factory
building, we decide the unit price as 237.57 €/m$^2$. Finally, the livestock house is still considered here
since two villages are included in the analysis, and the livestock house is built for sheep and cattle.
Therefore, the unit price of a livestock building is low (see Table 5). The economic loss of the buildings
in the Longxihe Basin are presented in Fig. 12.

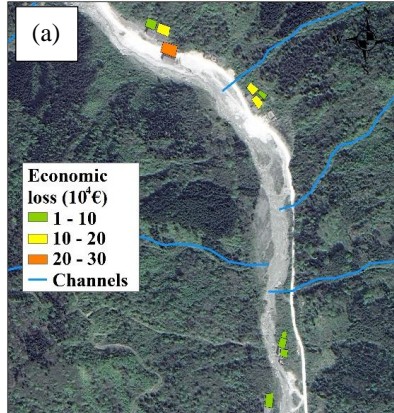
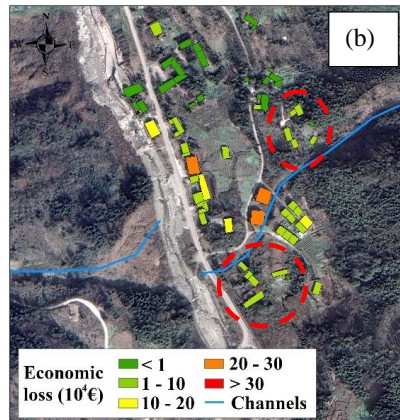

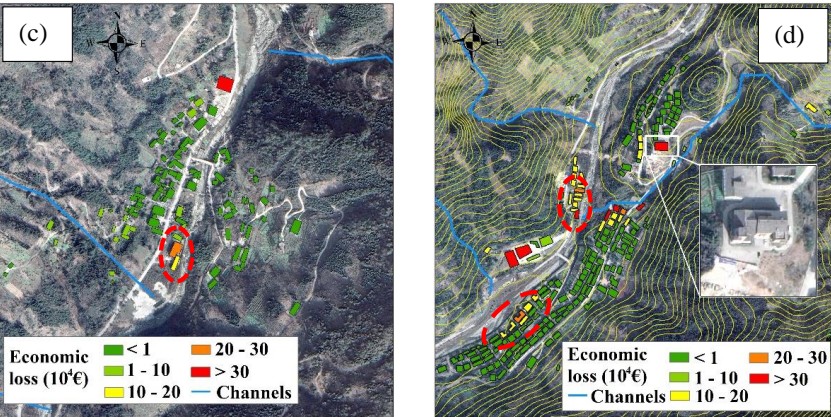

Figure 12. Economic loss of the residential buildings in the Longxihe Basin
The distribution characteristics of economic loss are different from physical vulnerability. For
example, Figs. 10(a) illustrates that the buildings are more likely to suffer severe damage if they are
close to the debris-flow channel, especially the non-RC frame structures. However, these non-RC frame
buildings require lower reconstruction or restoration costs when compared to the RC-frame buildings
(see Fig. 12(a)). In this case, the economic loss is low since it relies on the multiplication of physical
vulnerability and economic value of a building (see red dashes in Fig. 12(b)). As indicated in Fig. 12(d),
the factory buildings (see Fig. 10(d) and Fig. 12(d)) may suffer an economic loss of $3.2 \times 10^5 €$. As for
the reason why a low unit price of a factory building (see Table 5) results in a high economic loss may
be due to the large area of this factory building. Therefore, the site selection of a factory building is
significant. Although the location of the factory buildings in mountainous areas can avoid noise
pollution in urban development and decrease construction costs, the possible economic loss caused by
natural hazards cannot be neglected. Additionally, the residential building should not be built on the
outlet of the debris-flow catchment directly opposite (see red dash circles in Fig. 12(d)), especially when
the foundation of the residential buildings is only slightly higher than the riverway (see yellow contours
in Fig. 12(d)). For example, the highest economic loss is found in a residential building (see the image in
Fig. 12(d)), reaching $5.1 \times 10^5 €$. Therefore, at least a 4 m of residential building (RC frame) foundation





is essential if the buildings are close to the debris-flow channel based on Table 2. Overall, the analysis of
economic loss for buildings in mountainous areas can provide decision-makers with guidance about
urban planning.

**5. Discussion**

The proposed integrated method has been applied for the determination of the damage degree for

buildings in the Longxihe Basin, Sichuan, China. The involvement of debris-flow intensities, building
attributes, and spatial position between the buildings and debris-flow channel can help to suggest a more
reasonable damage degree value caused by debris flows. Specifically, the debris-flow intensity is
expressed in impact pressure here, which can indicate the possible consequence of a building if the
flowing materials strike the building directly. However, an overestimation of the damage degree may be
caused since the spatial positions between the building and debris-flow channel is not a one-dimensional
problem. In general, the elevation of a building is greater than that of the debris-flow channel in the
horizontal direction. This is because the long-term water flow and historical debris flows move the soils
and rocks, causing erosion of the channel bottom and therefore decreasing its elevation. As a result, the
elevation difference between the buildings and the debris-flow channel could cause a loss of impact
pressure. Therefore, simply utilizing impact pressure is not enough to reflect the actual damage to a
building. In contrast, the introduction of $HD$ and $VD$ is an effective supplement to improve the
estimation of physical damage that the buildings may suffer. Furthermore, the damage degree may vary
when subjected to different building structures. In this case, two major types of buildings are considered
in this study to distinguish the impact resistance capacities of different building types. Overall, this
developed matrix comprehensively describes the factors impacting the damage degree of buildings
caused by debris flows.

By utilizing the proposed matrix, we can estimate the damage degree of a building. However, the





possible damage in future scenarios is still unclear due to the change in debris-flow magnitude.
Therefore, an ensemble machine learning (ML) model is used to predict the volume of a future debris-
flow event so that the debris-flow intensities can be calculated based on the empirical relationships. This
ML method can effectively avoid over-fitting when training prediction models due to the existence of a
regular term. Most importantly, the strong ability in establishing a reliable relationship between a group
of independent variables and a dependent variable enables a wider application of ML methods when
compared to empirical and regression methods. Therefore, a precise prediction can be expected based on
the established relationship using the ML method to indicate the potential damage to buildings caused by
a future debris-flow event. However, we are also aware that the current sample size may not support a
robustness performance in estimating impact pressure to buildings. For broader applications, continuous
input of debris-flow data globally is essential, which may beyond the scope of this study. However,
further improvement can also be achieved if the floors of buildings are considered when developing the
physical vulnerability matrix. This is because the degree of loss presents a negative correlation with the
number of floors (Fuchs et al., 2019). Nevertheless, the limitation cannot alter the fact that our work can
benefit the subdivision of buildings in different vulnerability levels and provide suggestions about the
site selection of future residential areas.
**6. Conclusion**
In this paper, an integrated method for vulnerability assessment of buildings caused by future debris
flows was proposed. This method includes a physical matrix and a machine learning model, in which
this matrix was developed by considering the debris-flow process, building structure, and spatial
positions between the buildings and debris-flow channels. To be more specific, the debris-flow process
is represented by impact pressure ($P_t$), which can be estimated based on the debris-flow volume through
field investigations. As for the definition of spatial positions, *HD* and *VD* are used to describe the



position relation between the buildings and the debris-flow channel. By combining the three parameters, the actual impact pressure on the buildings can be decided. However, the damage degree may vary for different building structures. Therefore, the building structure is further considered to provide a precise estimation of the buildings, including the RC frame and non-RC frame (brick structure, light steel structure, and masonry structure). Therefore, a total of six types of buildings are included in this study. They are residential buildings (*RC* frame and brick structure), business buildings (*RC* frame), office buildings (*RC* frame), factory buildings (light steel structure), and livestock houses (brick structure). At the same time, an ML model (XGBoost) was developed to predict the impact pressure to buildings caused by future debris flows. On the basis of the proposed physical vulnerability matrix and machine learning model, we selected the Longxihe Basin, Sichuan, China, to conduct a case study. The results show that the non-RC buildings may be more likely to suffer severe damage if they are close to the debris-flow channels. The buildings with high and very-high physical vulnerabilities are mainly brick and light steel structures. Consequently, the factory buildings occupy the highest economic loss, reaching $2.41 \times 10^5$ € due to their large area. In addition, the buildings may suffer severe economic loss if they are located the directly opposite of the outlet of the debris-flow catchment. Overall, our studies can achieve a reliable assessment of the physical damage and corresponding economic loss of buildings and therefore provide suggestions and scientific support for the future construction planning of buildings.

**CRediT authorship contribution statement**

**Chenchen Qiu:** Methodology, Software, Data curation, Writing - Original draft preparation. **Xueyu Geng**: Conceptualization, Visualization, Validation, Supervision, Writing – Review & Editing.

**Declaration of competing interest**

The authors declare that they have no known competing financial interests or personal relationships that could have appeared to influence the work reported in this paper.

**Acknowledgement**

This work is financially supported by the European Union's Horizon 2020 research and innovation program Marie Skłodowska–Curie Actions Research and Innovation Staff Exchange (RISE) under grant





agreement [grant number 778360].
For the purpose of open access, the author has applied a Creative Commons Attribution (CC-BY)
licence to any Author Accepted Manuscript version arising from the submission.

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





**Appendix. A**
1. Mechanism of XGBoost
The mechanism of XGBoost is to constantly develop a new decision tree which acts as a weak
learner and fits the residual error of the last prediction. After the training of a total of $k$ trees, the final
prediction result is the sum of the score of each leaf node in each developed tree. The target function of
regression in XGBoost is:
$$L(\phi) = \sum_{i=1}^{m} l(y_i, y_i) + \sum_{k=1}^{t} \Omega(f_k) \qquad (10)$$

where $\sum_{i=1}^{m} l(y_i, y_i)$ represents the loss function, and $\sum_{k=1}^{t} \Omega(f_k)$ is the regularisation term. $y_i$ and $y_i$
are prediction value and true value, respectively. $m$ is the number of samples. $f_k$ is the $k_{th}$ tree model. As
mentioned above, the newly generated tree needs to fit the residual error of the last prediction, and
therefore the prediction result can be presented as:
$$y_i^t = y_i^{(t-1)} + f_t(x_i) \qquad (11)$$

Substitute the Eq. (12) into Eq. (11) to rewrite the objective function as:
$$L(\phi) = \sum_{i=1}^{m} l\left(y_i, y_i^{(t-1)} + f_t(x_i)\right) + \sum_{k=1}^{t} \Omega(f_k) \qquad (12)$$

Furthermore, Taylor's second order expansion is introduced to find $f_k$ to minimize the objective
function:
$$L(\phi) = \sum_{i=1}^{m} \left[ l\left(y_i, y_i^{(t-1)}\right) + g_i f_t(x_i) + \frac{1}{2} h_i f_t^2(x_i) \right] + \sum_{k=1}^{t} \Omega(f_k) + constant \qquad (13)$$

where $g_i$ is the first derivation, and the $h_i$ represents the second derivation
2. Calculation results of impact pressure $P_t$

Table 6. Dataset II for developing the impact pressure prediction model

| No. | $A$ (km²) | $L$ (km) | $R$ (m) | $J$ | $P_t$ (kPa) | No. | $A$ (km²) | $L$ (km) | $R$ (m) | $J$ | $P_t$ (kPa) |
|---|---|---|---|---|---|---|---|---|---|---|---|
| 1 | 8.55 | 3.13 | 269 | 0.1051 | 40.9 | 42 | 0.05 | 0.18 | 85 | 0.1908 | 10.3 |
| 2 | 4.68 | 1.41 | 126 | 0.2162 | 47.4 | 43 | 0.06 | 0.23 | 81 | 0.3038 | 14.2 |





| 3 | 12.88 | 4.16 | 269 | 0.1246 | 56.0 | 44 | 0.33 | 0.50 | 162 | 0.2792 | 18.4 |
|---|---|---|---|---|---|---|---|---|---|---|---|
| 4 | 0.29 | 0.50 | 95 | 0.1638 | 13.2 | 45 | 0.05 | 0.20 | 107 | 0.2661 | 12.2 |
| 5 | 0.29 | 0.29 | 200 | 0.4122 | 23.0 | 46 | 1.37 | 1.11 | 160 | 0.1763 | 34.1 |
| 6 | 5.73 | 0.71 | 260 | 0.1175 | 49.4 | 47 | 4.83 | 1.96 | 277 | 0.2071 | 35.5 |
| 7 | 0.56 | 0.62 | 195 | 0.2475 | 29.3 | 48 | 1.33 | 0.50 | 258 | 0.5117 | 35.7 |
| 8 | 2.15 | 0.73 | 250 | 0.2736 | 24.2 | 49 | 0.17 | 0.62 | 231 | 0.4727 | 21.0 |
| 9 | 0.32 | 0.46 | 276 | 0.5452 | 23.0 | 50 | 12.47 | 3.61 | 366 | 0.1853 | 67.9 |
| 10 | 1.67 | 0.95 | 161 | 0.3699 | 32.3 | 51 | 0.46 | 0.88 | 189 | 0.3819 | 26.4 |
| 11 | 11.21 | 1.93 | 360 | 0.1512 | 34.1 | 52 | 1.63 | 1.98 | 148 | 0.3115 | 28.9 |
| 12 | 2.85 | 1.57 | 232 | 0.2568 | 28.3 | 53 | 1.34 | 1.00 | 158 | 0.1727 | 18.7 |
| 13 | 2.29 | 1.84 | 189 | 0.3581 | 46.6 | 54 | 0.24 | 0.43 | 151 | 0.2867 | 16.6 |
| 14 | 0.08 | 0.42 | 240 | 0.3561 | 16.6 | 55 | 0.39 | 0.75 | 120 | 0.1745 | 15.6 |
| 15 | 0.18 | 0.48 | 366 | 0.6976 | 13.0 | 56 | 0.02 | 0.1 | 132 | 0.5295 | 18.0 |
| 16 | 0.53 | 0.81 | 170 | 0.2943 | 22.5 | 57 | 2.56 | 1.23 | 127 | 0.0998 | 16.7 |
| 17 | 0.71 | 1.74 | 151 | 0.6494 | 166.9 | 58 | 1.62 | 0.71 | 229 | 0.1673 | 19.7 |
| 18 | 0.49 | 1.64 | 162 | 0.6494 | 181.2 | 59 | 0.49 | 1.41 | 182 | 0.3000 | 24.0 |
| 19 | 0.60 | 1.52 | 155 | 0.6469 | 155.1 | 60 | 0.21 | 0.66 | 215 | 0.5384 | 40.6 |
| 20 | 0.36 | 1.15 | 261 | 0.8214 | 127.6 | 61 | 0.29 | 1.31 | 133 | 0.5184 | 64.1 |
| 21 | 2.73 | 2.57 | 190 | 0.6771 | 88.2 | 62 | 0.85 | 1.75 | 163 | 0.4578 | 36.0 |
| 22 | 2.02 | 2.59 | 198 | 0.7028 | 94.9 | 63 | 1.71 | 2.06 | 145 | 0.3879 | 68.5 |
| 23 | 0.43 | 1.30 | 198 | 0.7729 | 94.7 | 64 | 1.27 | 2.16 | 183 | 0.3522 | 84.1 |
| 24 | 0.19 | 1.09 | 181 | 0.6873 | 79.2 | 65 | 0.89 | 2.07 | 127 | 0.3385 | 68.1 |
| 25 | 1.03 | 2.02 | 232 | 0.4369 | 51.2 | 66 | 0.49 | 1.20 | 168 | 0.5681 | 141.0 |
| 26 | 3.99 | 3.78 | 134 | 0.4061 | 36.8 | 67 | 0.75 | 1.58 | 327 | 0.5566 | 165.7 |
| 27 | 2.88 | 2.40 | 313 | 0.7107 | 66.5 | 68 | 0.37 | 0.52 | 199 | 0.3404 | 23.6 |
| 28 | 0.34 | 1.14 | 163 | 0.8571 | 102.6 | 69 | 0.77 | 0.76 | 115 | 0.1566 | 17.0 |
| 29 | 2.81 | 2.84 | 253 | 0.5250 | 80.8 | 70 | 0.31 | 0.87 | 178 | 0.1317 | 25.9 |
| 30 | 7.18 | 4.82 | 400 | 0.5139 | 102.4 | 71 | 0.36 | 0.35 | 261 | 0.4578 | 20.6 |
| 31 | 24.42 | 9.47 | 337 | 0.3153 | 20.2 | 72 | 2.62 | 1.39 | 321 | 0.3482 | 33.8 |
| 32 | 2.81 | 1.74 | 205 | 0.3191 | 31.8 | 73 | 0.84 | 1.39 | 199 | 0.4899 | 14.9 |
| 33 | 0.43 | 1.30 | 200 | 0.8012 | 47.5 | 74 | 2.72 | 2.56 | 528 | 0.1069 | 31.2 |
| 34 | 7.06 | 4.41 | 275 | 0.4473 | 84.1 | 75 | 5.85 | 0.86 | 365 | 0.2962 | 31.5 |
| 35 | 1.07 | 2.05 | 225 | 0.4431 | 71.0 | 76 | 2.61 | 1.28 | 388 | 0.5317 | 44.0 |
| 36 | 0.86 | 2.17 | 149 | 0.3979 | 70.6 | 77 | 5.45 | 2.82 | 261 | 0.5228 | 112.0 |
| 37 | 6.51 | 2.92 | 252 | 0.5029 | 110.7 | 78 | 3.51 | 0.99 | 227 | 0.3839 | 38.2 |
| 38 | 0.42 | 1.64 | 151 | 0.4813 | 149.0 | 79 | 7.09 | 2.29 | 293 | 0.1962 | 52.6 |
| 39 | 0.51 | 1.43 | 153 | 0.4899 | 153.1 | 80 | 0.02 | 0.21 | 110 | 0.4390 | 17.8 |
| 40 | 0.20 | 0.76 | 130 | 0.5520 | 51.6 | 81 | 2.06 | 1.92 | 160 | 0.3211 | 29.7 |
| 41 | 0.34 | 1.25 | 130 | 0.4942 | 56.5 | | | | | | |

