# Peer review of "An integrated method for assessing vulnerability of buildings caused by debris flows"

_Natural Hazards and Earth System Sciences, 2024_

## Author Comment (AC1)

(1) Page 7, Eq. (1). It's better to define $V_p$ first before it appears in your manuscript.

*Answer: Thank you. We have revised it. Please see the 'Revised manuscript with changes marked'.*

(2) Page 8, line 115. Please be attention with your writing. It should be 'As suggested by Jakob et al. (2012) and Kang and Kim (2016)'.

*Answer: Thank you. We have revised it.*

(3) Page 8, line 122. I think it's better to add a reference to support your statement that 'the moderate slope gradient and frictional resistance would decrease the kinetic energy of travelling mass'.

*Answer: Thank you. We have added one citation to support our statement.*

(4) Page 8, line 129, Eq. (2). Please clarify what $h$ stands for? It is mean flow depth or the maximum flow depth? And also the debris-flow velocity, $v$.

*Answer: Thank you. h represents the deposit depth on buildings, and v is the flow velocity at the maximum discharge. We have added the definitions to Eq. (2) of the 'Revised manuscript with changes marked'.*

(5) Page 8, line 134. Same question that you need to define $Q_p$ and then use $Q_p$ to represent a physical parameter.

*Answer: Thank you. We have defined $Q_p$ before the introduction of Eq. (3).*

(6) Page 9, line 145. I think the final equation of $P_t$ is missing, which is represented by debris-flow volume and channel gradient.

*Answer: Thank you. We have provided the final equation of $P_t$ in section 2.3 since Pt still needs further development to link it to influence factors so that we can establish a prediction model to estimate potential impact pressure in the future.*

(7) Page 12, line 220. Please explain that how you estimate the impact pressure of 30 kPa to the damaged buildings? Additionally, how do the authors define that 'buildings stay undamaged'? Do you mean there is no damage to the building wall and the structure ? Do you involve the furniture or household appliances and equipment into your analysis since they also belong to properties.

*Answer: Thank you. We used Eq. (2) to estimate the impact pressure caused to buildings during field investigations because we h and v can be calculated easily based on the estimated deposition volume of a debris-flow event on the accumulation fan. $\rho_{df}$ is decided as 2000 kg/m³ in our study. In this case, we can estimate the impact pressure that buildings had suffered during an event. Additionally, we defined 'undamaged' as no damage of structure and walls. We agree with the statement that furniture or*

*household appliances also belong to properties. But in our study, we mainly focused on the damages of building structures, which occupies majority of possible economic loss. Of course, further studies are needed to involve furniture and household appliances into economic loss analysis.*

(8) Page 16, line 263. Is there a difference between the maximum flow depth ($h$) here and the flow depth in Eq. (2) ? Please be clear with the definitions in your manuscript.

*__Answer__: Thank you. They are different, and the maximum flow depth here indicates the flow depth that occurred during the flowing and deposition processes of an event. We have used $h_{max}$ to define this parameter. Please see line 263 of the 'Revised manuscript with changes marked'.*

(9) Page 16, Line 265. Please indicate that whether the height of building is involved into the elevation calculation.

*__Answer__: Thank you. The height of a building was not involved into calculations in our study. However, we are also aware that the different stories in a building may suffer different damages by a debris-flow event, which could promote the establishment of a more comprehensive and reliable evaluation matrix. Nevertheless, this cannot alter the fact that our proposed physical vulnerability matrix has further improved the vulnerability assessment of a building caused by a future debris-flow event. Of course, we will keep working on the improvement of this evaluation matrix to let it to be achievable of a more reliable damage evaluation.*

(10) Page 18, line 290. Please indicate the ratio of a training set and validation set.

*__Answer__: Thank you. We have clearly indicated the ratio of a training set and validation set. Please see the 'Revised manuscript with changes marked'.*

(11) Page 20, line 327. Please indicate the satellite image that you used to extract the buildings since readers need to know the resolution of your satellite images, which may cause uncertainty of physical vulnerability estimation.

*__Answer__: Thank you. As indicated in section 2.1.2, Gaofen-2 (GF-2) satellite images were used in this study to extract the buildings. This satellite can capture panchromatic (black and white) images with a spatial resolution reaching 0.8 m and multispectral (color) images with a spatial resolution up to 3.2 m. Therefore, the resolution of satellite images used for buildings identification is 0.8 m. We have clearly indicated the name of satellite used for building extractions. Please see the 'Revised manuscript with changes marked'.*

---

## Author Comment (AC2)

(1) Page 5 L82-84 "The formation of terrain in this area …" Please provide relevant references.

*Answer: Thank you. We have provided references to support this sentence. Please see the 'Revised manuscript with changes marked'.*

(2) Page 6 Figure 1. This flowchart outlines the core of your methodology and presents a significant amount of information. However, the current figure lacks clarity. For instance, what do the different colors represent? Additionally, terms such as the GridSearch algorithm, n_estimators, and learning_rate, etc., are not clearly explained, either in the figure caption or elsewhere in the manuscript.

*Answer: Thank you. We have improved this flowchart to clearly indicate the functions of different parts. In this figure, different colors represent the different steps in our proposed method. We have clearly indicated the optimal values of these parameters of XGBoost using GridSearch. Please see lines 219-222 of 'Revised manuscript with changes marked'.*

(3) Page 7 Figure 2. It would be clearer for readers if different colors were used to distinguish the debris flow events in each dataset.

*Answer: Thank you. We have used different colors to highlight the two datasets in Fig. 2.*

(4) Page 8 Eq. 3. How do you determine channel gradient J as it varies along the channel.

*Answer: Thank you. It's true that J changes along the channel. In our study, we focused on the mean gradient of the main channel within a debris-flow catchment, and it is calculated using the equation proposed by IMHE (1994):*

$$J = \frac{\left( \sum_{j=1}^{m} \left( E_{j-1} + E_j \right) L_j - 2E_0 L \right)}{L^2}$$

*where J is the mean path gradient (‰). $E_j$ (j=1, 2, …, j-1) represents the elevation of each break point in the movement path (m). Elevation was downloaded from the ASF website (https://search.asf.alaska.edu/#/)) that can provide DEM with a spatial resolution of 12.5 m. $L_j$ is length of each section within the movement path (m). m is the number of sections. $E_0$ is the elevation of the endpoint of mass movement (m), and L is the length of the travel path (m). The divided sections are presented as below.*

[Figure]

Figure 1. Segments of the travel path

We have added the calculation of J to section 2.1.1 of 'Revised manuscript with changes marked'.

(5) Page 9 L154. The elevation resolution for *VD* is also 0.8 m?

***Answer****: Thank you. Sorry for the confusion caused to you. The VD is decided based on the DEM (digital elevation model). We have indicated the calculation of VD in lines 68-70 of the 'Revised manuscript with changes marked'.*

(6) Page 12 Eq. 13 and Page 19 L296. How do you obtain the actual value of impact pressure for the calculation of MAPE?

***Answer:*** *Thank you. We first estimated the depositional volume of a debris-flow event during filed investigations based on the measured width, runout distance, and deposition depth on the accumulation fan. Then, the Eqs. (2)-(7) were used to calculate the impact pressure to buildings. After that, we established such an evaluation matrix on the basis of the exact damage of buildings, spatial locations between buildings and debris-flow channel, and building structures.*

(7) Page 20 L314. Where is the data source of the economic loss?

***Answer****: Thank you. We have added reference to support this data. Also, it can be found in the study of Ma et al. (2017). (Ma, Y. and Li, C., 2017, April. Research on the debris flow hazards after the Wenchuan earthquake in Bayi Gully, Longchi, Dujiangyan, Sichuan Province, China. In 2017 International Conference on Advanced Materials Science and Civil Engineering (AMSCE 2017) (pp. 166-170). Atlantis Press.)*

(8) Page 21 Figure 9. It would be helpful to indicate the precise locations of sub-figures a, b, d, and e within sub-figure c. Why is the color representing buildings brown in sub-figure c, while it is yellow in the other sub-figures?

***Answer****: Thank you. We have revised this figure, as you suggested. Please see Fig. 10 of the 'Revised manuscript with changes marked'.*

Technical corrections:

(9) Page 9 L150. Viscose debris flows should be viscous debris flows?

*Answer: Thank you so much for pointing out this mistake. We have revised it.*

(10) Page 12 L213. yire should be yipre?

*Answer: Thank you. We have revised it.*

---

## Author Comment (AC3)

Responses to comments of Reviewer 3

(1) Line 105: Please explicitly define damage degree. Is it the vulnerability matrix? A more explicit statement of that in the beginning of 2.1 would be clarifying. Line 155: Please elaborate on the fishnet tool and what it does briefly, beyond it's purpose for your methods

*Answer: Thank you. We have improved the definition of damage degree to let it to be more clear. Yes, it indicates the vulnerability matrix. As for the 'fishnet' tool in ArcGIS, it is used to create a grid of rectangular or square cells, forming a structure similar to net across a specified area. This grid, named 'fishnet', is usually used for spatial-related tasks, such as sampling, density calculations, and overlay analysis. Each cell in the grid is a polygon, and we can specify the cell size, and the extent of the grid. In our study, considering the difficulty in extracting single building, 'fishnet' tool was employed to generate rectangle cells from building clusters with specified cell size, 500 $m^2$ for a building.*

(2) Page 6, Figure 1: please explain variables and their names more explicitly, especially in the GridSearch algorithm section. Also, please indicate choice of colors and corresponding meanings

*Answer: Thank you. We have further explained the variables presented in Fig. 1. $V_p$ is the physical vulnerability of buildings, and $P_t$ represents the impact pressure of a debris-flow event to buildings. HD and VD are horizontal and vertical distance of buildings to their nearest debris-flow channel. These factors in dataset I are used to develop a physical vulnerability matrix. As for the variables in dataset II, they are the depositional volume of a debris-flow event (U), area of a debris-flow catchment (A), length of the main channel for a catchment (L), the average topographic relief (R), and the average gradient of main channel (J). $\rho_{df}$ is the mean density of the material. More details regarding the selection reason of these variables and their definitions have been provided in sections 2.1 and 2.3. Please see the 'Revised manuscript with changes marked'.*

*GridSearch algorithm was used to generate the optimal hyperparameters of XGBoost, including n_estimators, learning_rate, max_depth, min_child_weight, and gamma. Explanations of these hyper-parameters are provided as below.*

***n_esmators.*** *It determines the number of weak learners (decision tree) to be used in our model. The increased number of weak learners could improve accuracy but also*

*may result in overfitting.*

*   *learning_rate. This parameter controls the weight assigned to each decision tree added to the model. A small learning_rate can cause slow speed of model training, but it can improve the generalisation performance by requiring more decision trees (n_estimators), ultimately leading to a more robust model.*

*   *max_depth. It controls the maximum number of splits that each tree can have. Deeper trees are able to describe more complex relationships, but an overfitting may not be avoided. In contrast, A smaller depth may result in underfitting because the trees are too shallow to capture data features.*

*   *min_child_weight. This parameter decides the minimum amount of data that is required to create a new leaf node on a decision tree. A higher value allows the algorithm to be more conservative, reducing the risk of overfitting by requiring larger samples to form a split.*

*   *gamma. This parameter defines the minimum improvement in the model's accuracy required to make a split. If this improvement is less than the specified gamma, the split will not occur. A higher gamma value makes the algorithm more conservative capable of preventing overfitting.*

*   *The optimal values of these hyper-parameters have been provided in section 2.3. Please see the 'Revised manuscript with changes marked'.*

*   *As for the selection of colors in this figure, the first part with deep blue color represents 'Data collection', as shown in the improved Fig. 1. After that, 'Model development' highlighted with orange was conducted to propose a physical vulnerability matrix and develop an impact pressure estimation model. In the development of machine learning mode, the optimal hyperparameters of XGBoost were generated by GridSearch algorithm. After the determination of physical vulnerability that building may suffer when a future debris-flow event occurs, their economic loss can be estimated caused by an event (deep green). Finally, we applied this framework to the Longxihe Basin, China to conduct a case study (dark red) to suggest the potential economic loss this basin may encounter if debris flows occur in this area.*

(3) Figures 7- 12: Please offer a more descriptive caption of the figures.

*   ***Answer****: Thank you. We have improved the descriptions of these figures. Please see the 'Revised manuscript with changes marked'.*